# Deleterious Effects of *Cymbopogon nardus* (L.) Essential Oil on Life Cycle and Midgut of the Natural Predator *Ceraeochrysa claveri* (Navás, 1911) (Neuroptera: Chrysopidae)

**DOI:** 10.3390/insects14040367

**Published:** 2023-04-08

**Authors:** Karina Caballero-Gallardo, Elton Luiz Scudeler, Daniela Carvalho dos Santos, Elena E. Stashenko, Jesus Olivero-Verbel

**Affiliations:** 1Environmental and Computational Chemistry Group, School of Pharmaceutical Sciences, Zaragocilla Campus, University of Cartagena, Cartagena 130015, Colombia; 2Functional Toxicology Group, School of Pharmaceutical Sciences, Zaragocilla Campus, University of Cartagena, Cartagena 130014, Colombia; 3Laboratory of Insects, Department of Structural and Functional Biology, Institute of Biosciences of Botucatu, UNESP—São Paulo State University, Botucatu 18600-000, SP, Brazil; 4Chromatography Laboratory, Research Centre of Excellence, CENIVAM, Industrial University of Santander, Bucaramanga 680006, Colombia

**Keywords:** essential oil, insect, morphological alterations, histology

## Abstract

**Simple Summary:**

Essential oils (EOs) with biopesticide effects are key to sustainable food production. Although it is thought that they possess lower impacts on natural enemies of pests, there is little information on this subject. The objective of this study was to assess the effects of *Cymbopogon nardus* EO on the life cycle and midgut morphology of the predator *Ceraeochrysa claveri*. Insects treated with *C. nardus* showed changes in the different stages of the insect development, such as lack of cocoon formation, dead pupa inside the cocoon, and malformed adults. EO-induced damage included injuries in the midgut epithelium, displaying detachment of columnar cells and the development of epithelial folds. These results suggest that *C. nardus* oil induces alterations on the life cycle and development of *C. claveri*.

**Abstract:**

*Cymbopogon nardus* (citronella) essential oil (EO) has been widely used in the cosmetic and food industry due to its repellent and fumigant properties. The aim of this study was to evaluate its effects on the life cycle and midgut morphology of the natural predator *Ceraeochrysa claveri*. Larvae were fed on sugarcane borer eggs (*Diatraea saccharalis*) pretreated with citronella EO solutions (1–100 µg/mL in methanol, 5 s) or solvent and air-dried at room temperature for 30 min. Larval and pupal stage duration, the percentage of emergence of the insect, and malformed insects were recorded. One day after adults emerged from their cocoons, adult insects were used to obtain their midgut and analyzed using light microscopy. The chemical composition of *C. nardus* EO revealed that citronellal (25.3%), citronellol (17.9%), geraniol (11.6%), elemol (6.5%), δ-cadinone (3.6%), and germacrene D (3.4%) were the predominant compounds. Exposure to the EO produced a significant change in development duration for third instar and prepupa of the insect. The observed alterations in the lifecycle included prepupae with no cocoon formation, dead pupa inside the cocoon, and malformed adults. Several injuries in the midgut epithelium of exposed adults were registered, such as detachment of columnar cells leaving only swollen regenerative cells fixed on the basement membrane, and the formation of epithelial folds. In summary, these data suggest that *C. nardus* oil has adverse effects on the life cycle and midgut morphology of a beneficial predator.

## 1. Introduction

The resistance of insects to synthetic products has prompted scientists to find novel sources of insecticides with broad-spectrum activities. Since ancient times, plants and their metabolites, in particular essential oils (EOs), have been used as tools for a variety of applications, from the treatment of diseases to control unwanted organisms [1]. Essential oils are mixtures of volatile, organic chemicals, that play several roles in the protection and development of plants [2]. These EOs contain many different types of compounds, such as monoterpenes and sesquiterpenes [3], with one or two constituents often dominating their physiological action [4].

The genus *Cymbopogon*, belonging to the Poaceae family (Gramineae), has been one of the most studied plants as a source of potential biopesticides. The EOs from this genus have immense commercial value in cosmetic, personal use product, detergent, perfumery, and pharmaceutical industries [5]. Two of its most commonly found chemical components, geraniol and citral, are widely used as fragrances in cosmetics [6]. Moreover, *Cymbopogon* EO and its compounds possess many useful biological properties, including antiyeast, antibacterial, insecticidal, antifungal, and insect repellent activities [5].

Widely used EOs from the genus *Cymbopogon* have been isolated from key species, including palmarosa (*C. martinii*), lemongrass (*C. flexuosus* and *C. citratus*), and citronella (*C. nardus* and *C. winterianus*), all of them with extensive bibliography regarding their biological properties [7,8,9], among them their potential for the manufacture of biopesticides against pests such as *Tribolium castaneum* [10], *Cryptolestes* sp. [11], *Palorus subdepressus* [11], *Rhyzopertha dominica* [11], *Sitophilus zeamais* [11], *Anopheles funestus* [12], *Microcerotermes Beesoni* [13], and *Musca domestica* [14], among others.

A notorious species is *Cymbopogon nardus* Linn. Rendle (Poaceae), commonly known as citronella, is native to Thailand (Asia) [15], but is widely distributed in tropical and temperate regions of the world. This plant is a tall, perennial, fast-growing grass that contains a tuft of lemon-scented leaves from the annulate and sparingly branched rhizomes. The plant grows to a height of 1 m, from 5 to 10 mm in width, and has distinct bluish-green leaves that do not produce seeds. Its EO, citronella, is one of the most important to be derived from the Poaceae family, and its use has been reported in traditional medicine [16], displaying antifungal [17], antimicrobial [18], and insect repellent [19] characteristics, among others.

The EOs play a pivotal part of pest management strategies relying on their relative low mammalian toxicity, rapid biodegradation in the environment, and lower possibility of insect pest resistance, making them attractive alternatives to synthetic pesticides [20]. These volatile mixtures also exhibit a wide spectrum of biological actions against pest insects, acting as fumigants, contact insecticides, repellents, and antifeedants, and they may impact the growth rate, reproduction and behavior of these organisms [7,20,21].

A disadvantage of using synthetic chemicals is the low selectivity against non-target biota, causing harmful effects on natural enemies [22]. Beneficial insects are formidable predators of pests, thus reducing their populations through the use of chemicals in agricultural fields, and can indirectly lead to an increase in pest populations [23]. Alternatively, biological or chemical pressures on helpful species may also lead to their transformation in pests. In general, it is accepted that EOs and their components are usually less harmful to humans and animals than broad-spectrum synthetic pesticides. However, it is necessary to quantify their impact on non-target organisms, including natural pest controllers.

Green lacewings (Neuroptera: Chrysopidae) of Brazilian agro-ecosystems, also known as chrysopids, are important biological control agents of phytophagous arthropod populations in crops. These insects are predators found in several crops of economic interest, and they play diverse roles in the biological control of pests [24]. Within this family is *Ceraeochrysa claveri*, one of the most polyphagous predators commonly found in neotropical agroecosystems. Several studies have used the insect as a biological model to observe alterations in spermatogenesis [25], changes in larvae midgut cells [24], genome integrity [26], and cocoon spinning [27]. As *C. claveri* is a non-target, desirable species within the ecosystem, the main objective of this study was to assess the effects of citronella EO on its life cycle and midgut tissue structure.

## 2. Materials and Methods

### 2.1. Essential Oil Extraction

Aerial parts of *C. nardus* (Colombian National Herbarium, voucher number COL 578357) were collected from plants cultivated at the Research Centre of Excellence, CENIVAM, Industrial University of Santander, Bucaramanga, Colombia (07°08′44 N; 73°06′96 W; 977 m above sea level). The EO was isolated by microwave-assisted hydrodistillation (MWHD) of fresh leaves, and the identification of the EO components was carried out by gas chromatography coupled with mass spectrometry (GC-MS) according to previously reported methods [21].

### 2.2. Chemical Analysis of the C. nardus EO

The identification of the EO components was carried out by GC-MS as published elsewhere [21]. In brief, GC-MS was performed in a GC 7890 gas chromatograph (Agilent Technologies, AT, Palo Alto, CA, USA), equipped with a mass selective detector MSD 5975C (Electron ionization, EI, 70 eV; AT, Palo Alto, CA, USA), utilizing two capillary columns (DB-5MS, length 60 m × 0.25 mm internal diameter with 5% phenyl polydimethylsiloxane, PDMS 0.25 μm film thickness; and DB-WAX, length 60 m × 0.25 mm internal diameter with polyethylene glycol (PEG), 0.25 μm film thickness) with helium used as carrier gas at a flow rate of 1 mL/min. The components of the *C. nardus* EO were identified by retention times, linear retention indices (LRI), mass spectra interpretation, and comparison with databases (NIST, Wiley, Quadlib).

### 2.3. Insect Maintenance

Larvae of *C. claveri* (0–12 h old) employed in the experiments were gathered from the mass colony of the Laboratory of Insects in the Department of Morphology at the Institute of Biosciences of Botucatu at UNESP, Brazil. The *C. claveri* larvae were fed *Diatraea saccharalis* (Lepidoptera: Crambidae) eggs *ad libitum* until the pupal formation. Adults were reared on an artificial diet (1:1, *v*:*v*, honey/yeast mixture). The insects were kept in an environmental chamber under controlled conditions (25 ± 1 °C; 70 ± 10% RH; 12 L:12 D photoperiod) [28].

### 2.4. Insect Exposure to C. nardus EO

Fresh egg clusters of *D. saccharalis* (recently oviposited, 0–72 h old) were collected and dipped once in different EO solutions (0–100 µg/mL), using methanol as solvent, in a volume of 50 mL for 5 s, and then air-dried at room temperature for 30 min. In the control group, egg clusters were dipped in methanol. Newly hatched larvae were selected randomly and disposed in individual polyethylene pots (2 cm height × 6 cm diameter), which were divided into four experimental groups (n = 15 per group). The experiment groups were tested under the same environmental conditions as described for rearing. In the control group, larvae were fed *ad libitum* on *D. saccharalis* egg clusters treated with methanol. In the exposed groups (1, 10, and 100 µg/mL), larvae were fed *ad libitum* on eggs treated with *C. nardus* EO throughout the larval period until pupation. After cocoon spinning, specimens remained in the same recipients under the same controlled conditions, and until the adult emerged. Newly emerged adults were kept in a polyethylene box (9 cm height × 18 cm diameter) and fed the aforementioned artificial diet. One day after they emerged, insects were employed to isolate the midgut. Each experiment consisted of two replicates per concentration and control group (n = 30 per group), and it was repeated twice.

### 2.5. Effects of C. nardus Oil on Life Cycle Parameters

Larvae exposed to different concentrations of *C. nardus* EO (0–100 µg/mL) were used to evaluate the alterations in the life cycle of the insect. For life cycle parameters, the animals were checked daily to record the developmental time for larval instars. Mortality was recorded every day. The inability of prepupa to spin the cocoon to start the prepupal period, the pupal mortality inside the cocoon, the developmental time of the prepupa and the pupa, as well as malformed insects, were monitored daily following recommendations given by Scudeler et al. [28].

### 2.6. Light Microscopy

For each morphological study, at least five adults (n ≥ 3 from each experimental group) were processed and examined [24]. Animals were briefly cryoanesthetized and dissected in insect saline solution (0.1 M NaCl, 0.1 M Na_2_HPO_4_ and 0.1 M KH_2_PO_4_) under a stereomicroscope. The midguts were dissected and fixed in a solution of 2.5% glutaraldehyde and 4% paraformaldehyde in 0.1 M phosphate buffer (pH 7.3) for 24 h. After fixation, midguts were dehydrated through graded ethanol (70–95%), the midguts were embedded in glycol methacrylate historesin (Leica Historesin Embedding Kit, Leica Biosystems, Wetzlar, Germany), and sections of 3 µm were cut on Leica RM 2045 [24]. The midgut section slides were stained with 0.5% toluidine blue and 1% borax and analyzed and photographed with a Leica DM500 microscope (installed with LAS EZ-V2.0.0 software) using a ×40 objective.

### 2.7. Statistical Analysis

The data are expressed as the mean ± standard error of the mean. After checking for normality and variance homogeneity, using Shapiro–Wilk and Bartlett’s tests, respectively, ANOVA was used to evaluate the mean differences between the groups, utilizing Dunnett’s test as a post hoc test. When normality was not achieved, Kruskal–Wallis, followed by Dunn’s test, was utilized instead. The statistical tests were considered significant at *p* < 0.05.

## 3. Results

### 3.1. Characterization by GC-MS of the Essential Oil of C. nardus

Twenty-five compounds were identified in *C. nardus* EO, representing 95.8% of the total number of detected constituents. Major components (>10%) in the oil were citronellal [monoterpene] (25.3%), citronellol [monoterpene] (17.9%), and geraniol [terpene] (11.6%) as major components (>10%) (Table 1 and Figure 1). Other chemicals with noticeable levels in the mixture were elemol [sesquiterpene] (6.5%), δ-cadinene [sesquiterpene] (3.6%), limonene [monoterpene] (3.2%), and germacrene D [sesquiterpene] (3.4%).

### 3.2. Effects of C. nardus EO on the Life Cycle of C. claveri Insects

The *C. nardus* EO (1 and 10 µg/mL) exerted a significant decrease (15.7–22.1%) in the developmental time of the third instar larvae of *C. claveri* compared to the control. However, this parameter was prolonged when the exposure occurred at 100 µg/mL. In the prepupa stage, 1 and 100 µg/mL EO reduced the developmental time, whereas in the pupa stage the EO had no effect (Table 2).

The effects of *C. nardus* on pupation and pupal viability, as well as on morphological changes in adults of *C. claveri*, when larvae were fed the eggs of *D. saccharalis* treated with *C. nardus* EO, are presented in Table 3. The toxic effects on the prepupa and pupa were recorded at the highest concentration. Regarding anatomical changes, the control group displayed normal morphological features, without evident anatomical changes, featuring good development of antennae and wings, and complete separation from the cocoon (Figure 2A). However, adults exposed to *C. nardus* EO showed distinct morphological alterations when the dose increased, including a shrunken abdomen in freshly emerged adults (Figure 2B), no wing formation (Figure 2C), and inability to exit the cocoon (Figure 2D).

### 3.3. Histological Changes

The morphological changes observed in the midgut of adult *C. claveri* exposed to *C. nardus* EO are presented in Figure 3. Histological examination on the control individuals indicated that the midgut of adult *C. claveri* is defined by a pseudostratified epithelium composed of columnar and regenerative cells. These columnar cells feature spherical nuclei and striated borders. Regenerative cells are present in the basal area of the epithelium and normally occur in groups. Epithelial cells rest on a basement membrane and are wrapped externally by circular and longitudinal fibers of the gut musculature (Figure 3A). *C. claveri* adults exposed to *C. nardus* EO display serious injuries in the midgut epithelium, in particular detachment of columnar cells leaving only swollen regenerative cells fixed on the basement membrane, and the formation of epithelial folds (Figure 3B–D).

## 4. Discussion

The chemical composition of *Cymbopogon nardus* oil has been well documented, and the concentration of major components may vary to that published by other authors. For instance, citronellal, the major component may have concentrations that range from 22.2 to 37.8%, whereas citronellol may vary from 12.5 to 17.9% [29,30,31]. This variability is not surprising, as the composition of EOs depend on various factors such as light, precipitation, growing site, soil type, and genetic variability [32].

Chemicals present at high concentrations in *C. nardus* EO, in particular geraniol and citronellol, have been reported as highly efficient inducers of repellency against the booklouse, *Liposcelis bostrychophila*, and the red flour beetle, *Tribolium castaneum* [33]. In fact, this EO has great agricultural interest due to reports on its fumigant, repellent, and insecticidal action, properties that have also been evaluated on insects such as *Callosobruchus maculatus* [34], *Ulomoides dermestoides* [21], *Oryzaephilus surinamensis*, and *Sitophilus zeamais* [19]. As presented, the literature and data shown here with *C. claveri* evidence that *Cymbopogon nardus* EO has little specificity on its biotargets, and it may act on both natural predators and insects considered to be pests. Moreover, *Cymbopogon nardus* EO does not only target insects, its actions also include antifungal activity on maize fungi (*Fusarium verticillioides* and *Dreschlera maydis*) [29].

The effects elicited by *C. nardus* EO in the larval stage of *C. claveri* may be related to its known antifeedant and repellent properties. At low concentrations, the EO repelled fewer larvae from ingestion of the treated eggs, and therefore these larvae fed on large quantities of them, making them more susceptible to the harmful effects of the EO. By contrast, those exposed to the highest EO concentration (100 µg/mL) were strongly repellent to the larvae, but even the small amount of eggs consumed by this group resulted in lethal damage (no cocoon formation, death inside cocoon, and adult malformed). The same observations were found by Scudeler et al. [28] with neem oil.

In this study, a decrease (range: 7.7–23.1%) in viability of newly formed *C. claveri* adults was observed after exposure to *C. nardus* EO. At the greatest tested concentration, several effects appeared, involving changes in larval development and death, as also documented when *C. claveri* was exposed to neen oil (*Azadirachta indica*), where non-cocoon formation (17.3%) and dead insects inside the cocoon (10.5%) were observed [28].

The larvae exposed to *C. nardus* EO also developed important morphological changes in emerged adults. Although this has been documented for *C. claveri* adults after larval exposure to azadirachtin [26], a natural, wide-spectrum insecticide isolated from neem seeds, the frequency of the effects induced by the *C. nardus* EO was considerably lower. It is true that the neen oil is a biopesticide often preferred over synthetic ones, but similar to what has been shown here for *C. nardus* EO, it has the capacity to target predator, beneficial insects, as well [24,35].

After oral delivery, the gut is the first tissue in contact with the insecticide, and is divided into three zones (foregut, midgut, and hindgut). The use of insecticides causes changes in several parts of the insect, including the midgut [36]. This organ section is a region of digestion and absorption of nutrients modulated by the gut microbiota, housing one of the main defense mechanisms against pathogenic microorganisms [37]. Accordingly, the midgut is a target for biologically active chemicals, as it has been published for plant-derived molecules [38]. In fact, it has been reported that the midgut may be the site where most oral insecticide penetration occurs [39]. Ingestion of *C. nardus* EO by the insect seemed to affect the structure of its midgut epithelium, basically removing columnar cells, an adaptation feature that may result from impairment of digestion and absorption of nutrients. The effects of EOs on the midgut of insects have also been registered for *Cymbopogon winterianus* EO on *Spodoptera frugiperda* [40], causing morphological changes characterized by cytoplasmic protrusions, columnar cell extrusion, and pyknotic nuclei. In addition, *Melaleuca alternifolia* EO generated an elongation of digestive cells and necrosis on the midgut of *Podisus nigrispinus* [41], whereas exposure of *C. claveri* to neem oil induced cell dilation, cytoplasmic protrusions, and cell lysis in the same organ [24].

## 5. Conclusions

Exposure of *C. claveri* to *C. nardus* EO alters the duration its life cycle. Newly emerged *C. claveri* adults from larvae that ingested the EO exhibited incomplete development, as well as morphological alterations in the midgut structure of the insect, specifically absence of columnar cells and the formation of epithelial folds. The results found in this study suggest that the EO of *C. nardus* may be toxic to beneficial insects. These data encourage further investigations to assess the effects on other target and non-target organisms, contributing to a better understanding of EO in insects.

## Figures and Tables

**Figure 1 insects-14-00367-f001:**
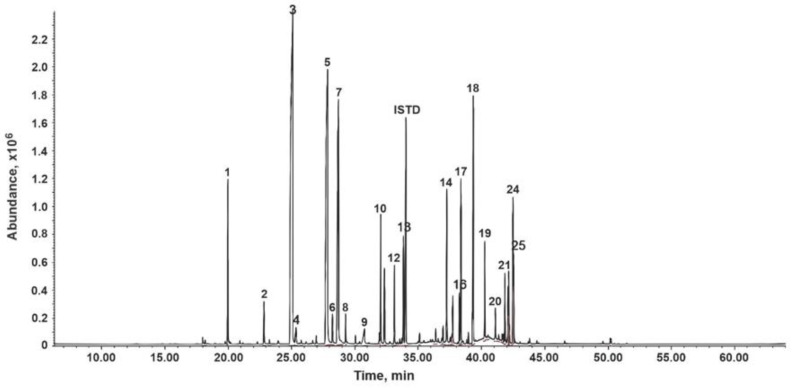
Chromatographic profile obtained by GC-MS (full scan) of *C. nardus* EO. Column DB-5MS (60 m), split injection 1:30, MSD (EI, 70 eV). Peaks: 1: Limonene, 2: Linalool, 3: Citronellal, 4: Pulegol, 5: Citronellol, 6: Neral, 7: Geraniol, 8: Geranial, 9: Citronellic acid, 10: Citronellic acid, 12: Geranyl acetate, 13: β-Elemene, 14: Germacrene D, 16: γ-Cadinene, 17: δ-Cadinene, 18: Elemol, 19: Germacrene D-4-ol, 20: C_15_H_26_O, 21: γ-Eudesmol, 24: α-Cadinol and 25: α-Eudesmol.

**Figure 2 insects-14-00367-f002:**
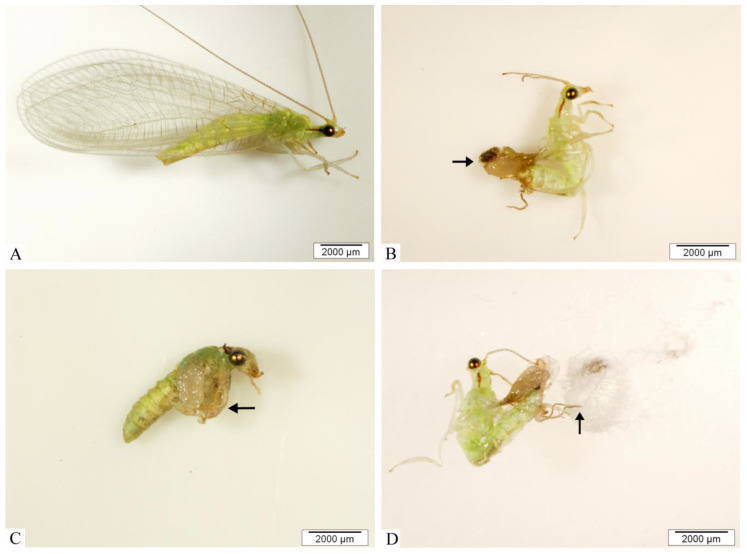
Macroscopic damage to adults of *C. claveri* after treatment with *C. nardus* oil. No deformity was observed in *C. claveri* adults (control group) (**A**), emergence and incomplete development following treatment with *C. nardus* (1 µg/mL, (**B**); 10 µg/mL, C and 100 µg/mL, (**D**)). (**C**) Note the non-formation and incomplete distension of the wings and antennae (*arrow*). (**D**) Note the inability of the adult to emerge from the cocoon (*arrow*). Photographs were taken with an Olympus SZX16 stereo microscope with Cell^D^ Imaging Software (Olympus Soft Imaging Solutions GmBH, Münster, Germany).

**Figure 3 insects-14-00367-f003:**
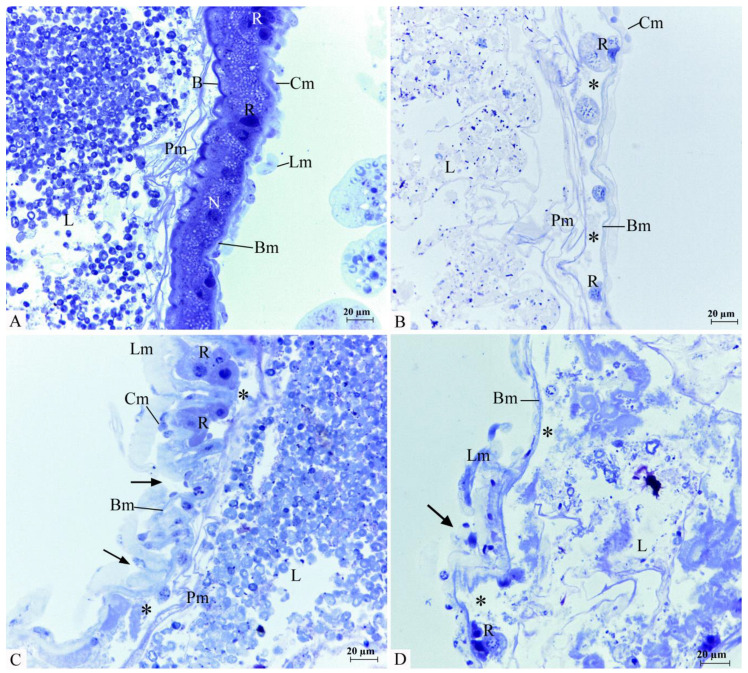
Morphological changes in the midgut epithelium of newly emerged adult *C. claveri* after treatment with *C. nardus* EO during the larval stage. (**A**) Control group. Epithelium is composed of columnar cells with a striated border (B) and nucleus (N) well developed and regenerative cells (R) in the basal area. These cells rest on a basement membrane (Bm) and are wrapped by circular (Cm) and longitudinal (Lm) fibers of the gut musculature. (**B**–**D**) *C. nardus* oil 1, 10, and 100 µg/mL, respectively. Note the absence of columnar cells (*) and the formation of epithelial folds (*arrows*). Perithrophic membrane (Pm); lumen (L).

**Table 1 insects-14-00367-t001:** Chemical composition of citronella EO obtained by GC-MS.

#	t_R_, min	Compound	Linear Retention Indices				
			DB-5MS		DB-WAX		Concentration %
			Exp.	Lit.	Exp.	Lit.	
1	19.98	Limonene	1032	1030 [2]	1194	1198 [2]	3.2
2	22.83	Linalool	1100	1099 [1]	1540	1543 [2]	0.8
3	25.09	Citronellal	1158	1154 [2]	1484	1475 [2]	25.3
4	25.35	Pulegol *	1165	1158 [3]	1623	1606 [3]	0.6
5	27.85	Citronellol	1231	1228 [2]	1762	1764 [2]	17.9
6	28.23	Neral	1242	1242 [2]	1674	1679 [2]	0.5
7	28.71	Geraniol	1255	1255 [2]	1842	1839 [2]	11.6
8	29.27	Geranial	1271	1270 [2]	1723	1725 [2]	0.5
9	30.76	Citronellic acid	1312	1312 [1]	2231	2232 [3]	0.8
10	32.04	Citronellyl acetate	1346	1350 [1]	1657	1657 [2]	2.7
11	32.35	Eugenol *	1355	1356 [1]	2144	2163 [2]	1.7
12	33.13	Geranyl acetate	1376	1380 [2]	1748	1751 [2]	1.7
13	33.85	β-Elemene	1395	1390 [2]	1590	1591 [2]	2.3
14	37.26	Germacrene D	1492	1484 [1]	1712	1708 [2]	3.4
15	37.74	α-Muurolene	1506	1500 [1]	1723	1723 [2]	1.1
16	38.26	γ-Cadinene	1523	1513 [1]	1762	1763 [2]	0.9
17	38.38	δ-Cadinene	1527	1523 [2]	1754	1756 [2]	3.6
18	39.36	Elemol	1559	1548 [1]	2073	2079 [2]	6.5
19	40.27	Germacrene D-4-ol	1588	1574 [1]	2043	2057 [2]	1.9
20	41.10	C_15_H_26_O	1617	-	-	-	0.6
21	41.85	γ-Eudesmol	1646	1631 [2]	2160	2176 [2]	1.3
22	42.09	*epi*-α-Cadinol	1653	1638 [2]	2176	2170 [2]	1.1
23	42.15	*epi*-α-Muurolol	1656	1641 [2]	2187	2186 [2]	1.4
24	42.49	α-Cadinol	1668	1652 [1]	2222	2227 [2]	2.8
25	42.55	α-Eudesmol	1670	1652 [1]	2220	2223 [2]	1.6

* Tentatively identified compound. [1] Adams, P. Identification of essential oil components by gas chromatography/mass spectrometry. 4th edición, Allured Publishing Corporation, Carol Stream, Illinois, 2004. [2] Babushok, V. I., Linstrom, P. J., Zenkevich, I. G. Retention Indices for Frequently Reported Compounds of Plant Essential Oils. J. Phys. Chem. 2011. 40 (4). 1–47. [3] NIST Mass Spectrometry Data Center. http://webbook.nist.gov/chemistry/ (accessed 21 December 2020).

**Table 2 insects-14-00367-t002:** Duration (days) of each phase in the life cycle of *C. claveri* exposed to *C. nardus* EO.

Treatment (µg/mL)	Developmental Time (Days)					
	First Instar	Second Instar	Third Instar	Prepupa	Pupa	Total Cycle
0	4.3 ± 0.1 (2–5) ^a^	3.6 ± 0.2 (3–6)	4.5 ± 0.2 (2–7)	5.0 ± 0.1 (4–6)	11.0 ± 0.2 (10–15)	28.4 ± 0.3 (26–32)
1	4.4 ± 0.2 (4–6)	4.0 ± 0.1 (3–5)	3.8 ± 0.2 * (3–6)	4.6 ± 0.1 * (4–5)	10.6 ± 0.1 (10–11)	27.3 ± 0.3 (25–31)
10	4.4 ± 0.1 (4–5)	3.9 ± 0.1 (3–5)	3.5 ± 0.2 * (2–6)	4.7 ± 0.1 (4–5)	10.4 ± 0.2 (8–12)	27.0 ± 0.2 * (26–30)
100	4.7 ± 0.2 (4–6)	3.7 ± 0.1 (3–4)	4.9 ± 0.2 (4–6)	4.4 ± 0.2 * (3–5)	11.1 ± 0.4 (10–15)	28.8 ± 0.4 (27–33)

^a^ Mean values ± standard error, the range is presented in parentheses. * Significant difference in means when compared to control group (Kruskal–Wallis and Dunn’s post hoc tests, *p* < 0.05).

**Table 3 insects-14-00367-t003:** The effects of exposure to *C. nardus* EO on viability (%) during the formation of *C. claveri* adults.

Treatment (µg/mL)	Prepupa (%)		Pupa (%)		Adult (%)	
	Viable	No Cocoon Formation	Viable	Death Inside Cocoon	Viable	Malformed
0	96.3 ± 0.0	3.7 ± 0.0	100.0 ± 0.0	0.0	100.0 ± 0.0	0.0
1	92.6 ± 0.3	7.4 ± 0.3	96.2 ± 3.8	3.8 ± 3.8	92.6 ± 0.3	7.4 ± 0.3
10	89.5 ± 3.8	10.5 ± 3.8	96.4 ± 3.6	3.6 ± 3.6	93.1 ± 0.2	6.9 ± 0.2
100	88.5 ± 3.8	11.5 ± 3.8	92.3 ± 0.0	7.7 ± 0.0	76.9 ± 0.0	23.1 ± 0.0

## Data Availability

All datasets used in this study can be provided by the authors upon reasonable request.

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
