# Peer review of "Deleterious Effects of Cymbopogon nardus (L.) Essential Oil on Life Cycle and Midgut of the Natural Predator Ceraeochrysa claveri (Navás, 1911) (Neuroptera: Chrysopidae)"

_insects, 2023, doi:10.3390/insects14040367_

Round 1
Reviewer 1 Report
In this work, effects on the life cycle and midgut morphology of Ceraeochrysa claveri exposed to C. nardus EO were evaluated. The study is interesting and provides new insights. The results suggest that C. nardus oil has adverse effects on the insect's life cycle and causes harm to its midgut. The study is interesting and provides new insights. But there are several questions on writing and experiment design. A few minor suggestions should be considered before publication. The major comments are as follows:
1. Line 133: Incomplete parentheses at the abbreviation of essential oils.
2. The introductory section provides separate information on essential oils and predators. The authors should emphasize the association between C. nardus EO and predators.
3. Table 2: Please explain the data within brackets in the notes.
4. The data in Table 3 are missing standard errors.
5. In the discussion section, the authors should emphasize the discussion of the results. For example, how to understand that 100 µg/mL EO has a smaller impact on the life cycle of predators. What is the correlation between predator life cycle and survival results.
Author Response
In this work, effects on the life cycle and midgut morphology of Ceraeochrysa claveri exposed to C. nardus EO were evaluated. The study is interesting and provides new insights. The results suggest that C. nardus oil has adverse effects on the insect's life cycle and causes harm to its midgut. The study is interesting and provides new insights. But there are several questions on writing and experiment design. A few minor suggestions should be considered before publication. The major comments are as follows:
Line 133: Incomplete parentheses at the abbreviation of essential oils.
Answer
Thanks for your suggestion. The name of the compound has been corrected.
The introductory section provides separate information on essential oils and predators. The authors should emphasize the association between C. nardus EO and predators.
Answer
Thanks for your comment. The section of the introduction has been modified according to the suggestion.
Table 2: Please explain the data within brackets in the notes.
Answer
Thanks for your suggestion. The data within brackets have been explained.
The data in Table 3 are missing standard errors.
Answer
Thanks for your suggestion. The standard error has been added to the data in the Table 3
In the discussion section, the authors should emphasize the discussion of the results. For example, how to understand that 100 µg/mL EO has a smaller impact on the life cycle of predators. What is the correlation between predator life cycle and survival results.
Answer: Thanks for your suggestion. The following paragraph has been included in the discussion section “In this work, the antifeedant and repellent effects of the C. nardus EO could explain the effects elicited in the larval stage. At low concentrations, the EO repelled fewer larvae from ingestion of the treated eggs, and therefore these larvae fed on large quantities of them and became more susceptible to the harmful effects of the EO. By contrast, those ex-posed to the highest EO concentration (100 µg/mL) were strongly repellent to the larvae, but even the small amount of eggs consumed by this group resulted in lethal damage (no cocoon formation, death inside cocoon and adult malformed), the same observations were found by Scudeler et al. [35] with neem oil.”
Reviewer 2 Report
The study examined the toxicity of Cymbopogon EO on the Chrysopidae C. nardus, which is an interesting topic since the impact of such EO on natural enemies has often been overlooked. However, the manuscript has several weaknesses, including a low number of replications, poor English style, and a lack of references to past studies on similar topics.
The study did not find citral, one of the main components of Cymbopogon spp EO, which has been reported in the literature.
The discussion should focus more on why the information in the study is important.
The introduction is too brief and should provide more information on why the study was conducted. The importance of Chrysopidae as natural enemies of herbivores should be emphasized, as should the fact that Cymbopogon spp EO has already been tested as a candidate biocide on different insects. Several references on this topic have been overlooked:
Bossou, A. D., Ahoussi, E., Ruysbergh, E., Adams, A., Smagghe, G., De Kimpe, N., ... & Mangelinckx, S. (2015). Characterization of volatile compounds from three Cymbopogon species and Eucalyptus citriodora from Benin and their insecticidal activities against Tribolium castaneum. Industrial Crops and Products, 76, 306-317.
Doumbia, M., Yoboue, K., Kouamé, L. K., Coffi, K., Kra, D. K., Kwadjo, K. E., ... & Dagnogo, M. (2014). Toxicity of Cymbopogon nardus (Glumales: Poacea) against four stored food products insect pests. International Journal of Farming and Allied Sciences, 3(8), 903-909.
Ntonga, P. A., Baldovini, N., Mouray, E., Mambu, L., Belong, P., & Grellier, P. (2014). Activity of Ocimum basilicum, Ocimum canum, and Cymbopogon citratus essential oils against Plasmodium falciparum and mature-stage larvae of Anopheles funestus ss. Parasite, 21.
Olotuah, O. F., & Dawodu, E. O. (2017). Biocidal Properties of Cymbopogon Citratus Extracts on Termite (Microcerotermes Beesoni). Agriculture and Food Sciences Research, 4(1), 20-23.
Samarasekera, R., Kalhari, K. S., & Weerasinghe, I. S. (2006). Insecticidal activity of essential oils of Ceylon Cinnamomum and Cymbopogon species against Musca domestica. Journal of essential oil research, 18(3), 352-354.
Line 53, revise the sentence “…essential oils are used by insects for communication…” it is wrong concept and badly expressed.
Line 57, genera of what?
Lines 115-118, the number of replication carried out is very low.
Table 1, revise “linalool”, revise “tr” in RT ansd specify what is in the caption. Add the compound that were identified by standard. I understand that the study was carried out using an existing methodology, but it should be summarized in MM in brief.
In discussion, first summarize the main findings of the study, the lines 218-225 should be reported in introduction or later in the discussion.
Line 229, this was stated before, delete
Line 237, revise this sentence. Delete “…compounds are classified as…”
Author Response
The study examined the toxicity of Cymbopogon EO on the Chrysopidae C. nardus, which is an interesting topic since the impact of such EO on natural enemies has often been overlooked. However, the manuscript has several weaknesses, including a low number of replications, poor English style, and a lack of references to past studies on similar topics.
Answer
Thanks for your comments. We tried to answer all of them as well as we could.
The study did not find citral, one of the main components of Cymbopogon spp EO, which has been reported in the literature.
Answer
Citral did appear in the composition. It is a mixture of Neral (Compound 6) and Geranial (Compound 8).
The discussion should focus more on why the information in the study is important.
Answer
Thanks for your comment. The Discussion Section has been corrected following the referee´s suggestion.
The introduction is too brief and should provide more information on why the study was conducted. The importance of Chrysopidae as natural enemies of herbivores should be emphasized, as should the fact that Cymbopogon spp EO has already been tested as a candidate biocide on different insects. Several references on this topic have been overlooked:
Bossou, A. D., Ahoussi, E., Ruysbergh, E., Adams, A., Smagghe, G., De Kimpe, N., ... & Mangelinckx, S. (2015). Characterization of volatile compounds from three Cymbopogon species and Eucalyptus citriodora from Benin and their insecticidal activities against Tribolium castaneum. Industrial Crops and Products, 76, 306-317.
Doumbia, M., Yoboue, K., Kouamé, L. K., Coffi, K., Kra, D. K., Kwadjo, K. E., ... & Dagnogo, M. (2014). Toxicity of Cymbopogon nardus (Glumales: Poacea) against four stored food products insect pests. International Journal of Farming and Allied Sciences, 3(8), 903-909.
Ntonga, P. A., Baldovini, N., Mouray, E., Mambu, L., Belong, P., & Grellier, P. (2014). Activity of Ocimum basilicum, Ocimum canum, and Cymbopogon citratus essential oils against Plasmodium falciparum and mature-stage larvae of Anopheles funestus ss. Parasite, 21.
Olotuah, O. F., & Dawodu, E. O. (2017). Biocidal Properties of Cymbopogon Citratus Extracts on Termite (Microcerotermes Beesoni). Agriculture and Food Sciences Research, 4(1), 20-23.
Samarasekera, R., Kalhari, K. S., & Weerasinghe, I. S. (2006). Insecticidal activity of essential oils of Ceylon Cinnamomum and Cymbopogon species against Musca domestica. Journal of essential oil research, 18(3), 352-354.
Answer
Thanks for your comment.
The Introduction Section has been modified following the Referee´s recommendations. Suggested references were added to this Section.
Line 53, revise the sentence “…essential oils are used by insects for communication…” it is wrong concept and badly expressed.
Answer
Thanks for your suggestion. The sentence has been deleted.
Line 57, genera of what?
Answer
Thanks for your suggestion.
The sentence has been corrected.
Lines 115-118, the number of replication carried out is very low.
Answer
The total number of insects selected according to C. nardus EO concentration assayed was added.
Table 1, revise “linalool”, revise “tr” in RT and specify what is in the caption. Add the compound that were identified by standard. I understand that the study was carried out using an existing methodology, but it should be summarized in MM in brief.
Answer
A general description of the methodology for the characterization of EO has been included in the materials and methods section.
The names of the peaks have been added in the legend of Figure 1.
In discussion, first summarize the main findings of the study, the lines 218-225 should be reported in introduction or later in the discussion.
Answer
Thanks for your comment. The section of the discussion has been corrected according to the suggested.
Line 229, this was stated before, delete
Answer
Thanks for your suggestion. The sentence has been deleted.
Line 237, revise this sentence. Delete “…compounds are classified as…”
Answer
Thanks for your suggestion. The sentence has been corrected.
Round 2
Reviewer 2 Report
Overall, while the manuscript has undergone some improvements, it still falls short in addressing the most crucial point. Specifically, the toxicity of the EO towards the natural enemy used in the study renders it an unsuitable candidate for use as an alternative to pesticides. This finding constitutes the main result of the study and should be given more attention in the discussion section.
Moving forward, it is recommended that the entire manuscript be revised to focus on the fact that EO's are commonly reported as being less toxic to natural enemies, whereas in reality they can be harmful to predators as C. nardus. Additionally, the English language used in the manuscript requires significant improvement and should be reviewed by a native-speaking entomologist.
Taking these points into account, it is strongly advised that the manuscript undergo major revisions.
Line 25, delete “in short”
Line 31, it is not stated before that citronella is Cymbopogon nardus, but only later on at line 74.
line 69, delete “and”
line 98, revise
in table “linalool” should be corrected in “linalool”
across the manuscript the names of the species families and the person who described first the species is never reported.
Lines 268-269, bad English style
Lines 273-274, unclear meaning.
Line 275, what is a “slightly higher results were found in…”?
Line 277, author mean dead individuals?
Lines 311-313 this is untrue, because the authors didn’t test the EO toward a pest but versus a natural enemy of herbivores, so the only data obtained from the study is the EO toxicity values toward C. claveri, and unfortunately are negative results.
Author Response
Reviewer 2
Overall, while the manuscript has undergone some improvements, it still falls short in addressing the most crucial point. Specifically, the toxicity of the EO towards the natural enemy used in the study renders it an unsuitable candidate for use as an alternative to pesticides. This finding constitutes the main result of the study and should be given more attention in the discussion section. Moving forward, it is recommended that the entire manuscript be revised to focus on the fact that EO's are commonly reported as being less toxic to natural enemies, whereas in reality they can be harmful to predators as C. nardus. Additionally, the English language used in the manuscript requires significant improvement and should be reviewed by a native-speaking entomologist. Taking these points into account, it is strongly advised that the manuscript undergo major revisions.
Thanks for your suggestion. English language has been revised. The manuscript has been reviewed in detail once more and adjusted
Line 25, delete “in short”
Answer: Thanks for your suggestion. The text “in short” has been deleted.
Line 31, it is not stated before that citronella is Cymbopogon nardus, but only later on at line 74.
Answer: Thanks for your suggestion. The text has been arranged according to your suggestion.
Line 69, delete “and”
Answer: Thanks for your suggestion. The word “and” has been deleted.
Line 98, revise
in table “linalool” should be corrected in “linalool”
Answer: Thanks for your suggestion. The name “linalol” has been corrected.
across the manuscript the names of the species families and the person who described first the species is never reported.
Answer: Thanks for your suggestion. The information has been included in the manuscript.
Lines 268-269, bad English style
Answer: Thanks for your suggestion. These lines have been revised and corrected.
Lines 273-274, unclear meaning.
Answer: Thanks for your suggestion. These lines have been revised and corrected.
Line 275, what is a “slightly higher results were found in…”?
Answer: Thanks for your suggestion. These lines have been revised and corrected.
Line 277, author mean dead individuals?
Answer: Yes, these are dead individuals. This was corrected.
Lines 311-313 this is untrue, because the authors didn’t test the EO toward a pest but versus a natural enemy of herbivores, so the only data obtained from the study is the EO toxicity values toward C. claveri, and unfortunately are negative results.
Answer: Thanks for your suggestion. These lines have been revised and corrected.